# THE FAIRNESS-ACCURACY LANDSCAPE OF NEURAL CLASSIFIERS

## ABSTRACT

That machine learning algorithms can demonstrate bias is well-documented by now. This work confronts the challenge of bias mitigation in fully-connected feedforward neural nets through the lens of multiobjective optimisation. Specifically, recognising that fairness and accuracy are competing objectives, the proposed methodology uses techniques from multiobjective optimisation to ascertain the fairness-accuracy landscape of a neural net classifier. Along the way, a new causal notion of fairness is introduced that is particularly suited to giving a nuanced treatment when data is collected under unfair practices. In particular, special attention is paid to subjects whose covariates could appear with substantial probability in either value of the sensitive attribute. Experimental results suggest that the proposed method produces neural net classifiers that distribute evenly across the Pareto front of the fairness-accuracy space and is more efficient at finding non-dominated points than an adversarial approach.

## 1 INTRODUCTION

There is increasing concern over the ethics of machine learning algorithms. The issue of machine bias was prominently featured in ProPublica's 2016 eponymous article (Angwin et al., 2016) where the investigation uncovered prejudice against African-Americans in COMPAS (Correctional Offender Management Profiling for Alternative Sanctions), a recidivism prediction tool developed by Northpointe. Efforts to mitigate machine bias has received steadily increasing attention from stakeholders in a wide array of arenas including academia, industry research labs, and advocacy groups.

The question of fairness has a long history (Sen & McMurrin, 1980) and its definition continues to be debated (Dwork et al., 2012; Chouldechova, 2016; Joseph et al., 2016). For instance, Corbett-Davies et al. (2017) and Dieterich et al. (2016) argue that COMPAS can be regarded as fair with respect to certain fairness notions. Works such as Hardt et al. (2016) and Kleinberg (2018) have investigated this type of ambiguity by showing that certain fairness criteria cannot be simultaneously satisfied.

Having fixed a definition of fairness, most works in algorithmic fairness proceed by imposing constraints during the training process in an attempt to enforce fairness (Hardt et al., 2016; Joseph et al., 2016; Zafar et al., 2017a;b). Recent works can handle very complex learning algorithms such as neural networks (Beutel et al., 2017; Wadsworth et al., 2018; Madras et al., 2018; Manisha & Gujar, 2018). However none of the works in the literature seek to provide a holistic view of the fairness-accuracy landscape of the algorithm.

What we mean is this – though it is desirable that the neural network maintain high predictive accuracy while simultaneously remaining fair with regards to a sensitive variable such as race or gender, these two objectives often compete. Given this, it is helpful to cast the balancing act between fairness and accuracy as a multiobjective optimisation task and look at the fairness-accuracy *Pareto front*. This can give a bird's eye view of the algorithm and can be useful for comparing two algorithms based on, say, the "volume" of the Pareto front (Li et al., 2015). In this work, we propose a methodlogy for estimating the fairness-accuracy Pareto front of a fully-connected feedforward network.

**Contributions** This is the first work in algorithmic fairness that specifically addresses estimation of the fairness-accuracy Pareto front of a fully-connected feedforward network. The framework presented is flexible enough to allow for user-supplied accuracy and fairness measures. Along the

way, we introduce a causal measure of fairness which emphasises fairness amongst subjects with the most overlap in observed covariates across the different values of a sensitive attribute. Finally, the methodology can be used to enforce fairness in all intermediate representations of the neural network. This has potential benefits for downstream transfer learning tasks.

## 2 RELATED WORK

In this section, we review the broad categories of existing methods in algorithmic fairness. The major discernible classes of fair learning methods can be trifurcated according to the stage in which action is taken. The first class of methods attempts to remove bias from the input data itself. These methods rest on the premise that once proper preprocessing is accomplished, any classifier can be used to subsequently produce fair predictions (Kamiran & Calders, 2012; Feldman et al., 2015; Calmon et al., 2017; Johndrow & Lum, 2019).

On the other hand, post-processing techniques directly operate on the classifier output and are amenable to any classifier. The technique in Hardt et al. (2016) for instance seeks to learn a monotone transformation of the classifier's output to remove unfairness with regard to either demographic parity or equalised odds.

The third type of algorithmic fairness method directly intervenes in the stage of training. Many of these methods are specific to certain classifiers and certain notions of fairness. Generally speaking, train-time methods minimise predictive error while enforcing some fairness constraint (Calders & Verwer, 2010; Kamishima et al., 2011; Zafar et al., 2017a;b;c; Bechavod & Ligett, 2017; Agarwal et al., 2018; Narasimhan, 2018).

The proposed methodology falls into this category. However, rather than placing fairness constraints on the output of the classifier, our method nudge internal representations in the neural network to be less biased. It is akin to performing a sequence of supervised pre-processing to the input data, one in each layer of the neural net.

We will later implement in Section 6 an adversarial alternative to estimating the fairness-accuracy Pareto front. This is also a train-time algorithmic fairness method which employs concepts from adversarial learning. The main classifier is engaged in a game with an adversary adversary that tries to predict the sensitive attribute from the output of the predictor (Ganin et al., 2016; Beutel et al., 2017; Wadsworth et al., 2018; Zhang et al., 2018).

## 3 THE FAIRNESS-ACCURACY PARETO FRONT

This section introduces the fairness-accuracy Pareto front of a general learning algorithm. Suppose the inputs live in some space $\mathbb{X}$, the sensitive attribute (e.g. race or gender) in $\mathbb{A}$, and the response in $\mathbb{Y}$. Let $(\mathbb{X}, \mathbb{A}, \mathbb{Y})$ be a measurable space and $P$ be a probability measure on it. Let $\mathcal{F}$ be a class of functions from $\mathbb{X}$ to $\mathbb{Y}$. Given a loss function $\mathcal{L} : \mathbb{Y} \times \mathbb{Y} \to \mathbb{R}$, we may define the risk, $R(f; P) = \mathbb{E}_{(\mathbf{x}, \mathbf{a}, \mathbf{y}) \sim P} \mathcal{L}(f(\mathbf{x}), \mathbf{y})$.

Suppose $\mathcal{F}$ is chosen to be the family of functions $f_\theta : \mathbb{X} \to \mathbb{Y}$ parametrised by a fully-connected feedforward neural network with parameter $\theta \in \Theta$. For a probability measure $P$ on $(\mathbb{X}, \mathbb{A}, \mathbb{Y})$, define $R(\theta; P) = R(f_\theta; P)$. Let $U(\theta; P)$ be a measure of the *unfairness* of $f_\theta$, in a manner to be made precise in Section 4. Since we wish for the learning algorithm $f_\theta$ to be both accurate and fair, we wish to minimise, over $\theta$, the *vector* objective function

$$\begin{bmatrix} R(\theta; P) \\ U(\theta; P) \end{bmatrix}. \tag{1}$$

The optimisation is made difficult by the competing nature of the individual components. Take for instance as one extreme of the spectrum – performing classification completely at random. Then the resulting classifier will certainly be fair with respect to the sensitive attribute, by almost all measures of fairness. The other extreme might be to obtain a perfect classifier. But for datasets collected in a biased way, this will result in the classifier being unfair. For instance, a recidivism dataset would contain a disproportionate number of re-offenses amongst some group if police routinely target that group.

When the individual components of a vector objective compete, as they do in equation 1, it is un-likely that a parameter value exists which simultaneously minimises the individual objectives. This lack of total ordering necessitates optimisation according to a *partial order*. For $a, b \in \mathbb{R}^p$, we say $a \leq b$ if and only if every component of $a$ is less than or equal to the corresponding component of $b$. Suppose we have $p$ objective functions $J_1, \ldots, J_p$ where each is a function from the parameter space $\Theta$ to $\mathbb{R}$. Then $\theta \in \Theta$ is *Pareto optimal* if and only if there does not exist any $\tilde{\theta} \in \Theta$ such that $(J_1(\tilde{\theta}), \ldots, J_p(\tilde{\theta})) \leq (J_1(\theta), \ldots, J_p(\theta))$ with at least one strict inequality. The **Pareto front** is the set of all Pareto optimal points.

A basic technique for approximating the Pareto front is to scalarise the vector objective function. Let $\lambda \in [0, 1]$. One possible scalarisation scheme for equation 1 forms the convex combination $(1 - \lambda)R(\theta; P) + \lambda U(\theta; P)$. An important caveat is that scalarisation in this manner only allows for recovery of points on the *convex hull* of the Pareto front (Das & Dennis, 1997). A scalarisa-tion scheme that avoids this issue is the so-called Chebyshev method (Ehrgott, 2000; Giagkiozis & Fleming, 2015) in which we take

$$\theta^\lambda = \arg\min_\theta \ \max\{(1 - \lambda)R(\theta; P), \lambda U(\theta; P)\}. \tag{2}$$

The Chebyshev scalarisation enjoys several desirable properties. It guarantees solutions that are at least *weakly Pareto optimal* for any $\lambda \in [0, 1]$. The term weakly refers to replacing the non-strict inequality in the Pareto optimal definition with a *strict* inequality. A further property of the Chebyshev scalarisation is that any Pareto optimal solution can be obtained for some $\lambda$.

**Estimation of the Pareto front** Now we describe a general technique for estimating the set $\{\theta^\lambda : \lambda \in [0, 1]\}$ where $\theta^\lambda$ is given by equation 2. Let $(\mathbf{x}_1, \mathbf{a}_1, \mathbf{y}_1), \ldots, (\mathbf{x}_n, \mathbf{a}_n, \mathbf{y}_n)$ be inde-pendent copies of $(\mathbf{x}, \mathbf{a}, \mathbf{y})$ drawn from (unknown) distribution $P_{\text{model}}$. Define the empirical mea-sure as $\hat{P}_{\text{data}} = \frac{1}{n}\sum_{i=1}^n \delta_{(\mathbf{x}_i, \mathbf{a}_i, \mathbf{y}_i)}$. Let $R(\theta; \hat{P}_{\text{data}})$ be the plug-in estimator of $R(\theta; P_{\text{model}})$, i.e. $R(\theta; \hat{P}_{\text{data}}) = \mathbb{E}_{(\mathbf{x}, \mathbf{a}, \mathbf{y}) \sim \hat{P}_{\text{data}}} \mathcal{L}(f(\mathbf{x}), \mathbf{y}) = \frac{1}{n}\sum_{i=1}^n \mathcal{L}(f(\mathbf{x}_i), \mathbf{y}_i)$. Let $U(\theta; \hat{P}_{\text{data}})$ be an estimate of $U(\theta; P_{\text{model}})$, but not necessarily a plug-in estimator. Then we consider the empirical version of equation 2 given by

$$\hat{\theta}_n^\lambda = \arg\min_\theta \max\{(1 - \lambda)R(\theta; \hat{P}_{\text{data}}), \lambda U(\theta; \hat{P}_{\text{data}})\}. \tag{3}$$

**Evaluation** To assess the quality of our Pareto front approximation, it will be helpful to have unbiased estimators of $R(\hat{\theta}_n^\lambda; P_{\text{model}})$ and $U(\hat{\theta}_n^\lambda; P_{\text{model}})$. Suppose we have a test set $\mathbb{V} = \{(\mathbf{x}_i^*, \mathbf{a}_i^*, \mathbf{y}_i^*)\}$ where $(\mathbf{x}_1^*, \mathbf{a}_1^* \mathbf{y}_1^*), \ldots, (\mathbf{x}_m^*, \mathbf{a}_m^*, \mathbf{y}_m^*)$ are another set of independent copies of $(\mathbf{x}, \mathbf{a}, \mathbf{y})$ drawn from distribution $P_{\text{model}}$. Define the corresponding empirical measure as $\hat{P}_{\text{test}} = \frac{1}{m}\sum_{i=1}^m \delta_{(\mathbf{x}_i^*, \mathbf{a}_i^*, \mathbf{y}_i^*)}$. The risk of $\hat{\theta}_n^\lambda$ can be assessed using the out-of-sample average loss $R(\hat{\theta}_n^\lambda; \hat{P}_{\text{test}}) = \mathbb{E}_{(\mathbf{x}, \mathbf{a}, \mathbf{y}) \sim \hat{P}_{\text{test}}} \mathcal{L}(f(\mathbf{x}; \hat{\theta}_n^\lambda), \mathbf{y}) = \frac{1}{m}\sum_{i=1}^m \mathcal{L}(f(\mathbf{x}_i^*; \hat{\theta}_n^\lambda), \mathbf{y}_i^*)$. The unfairness can be also assessed on the test set, let's denote it $U(\hat{\theta}_n^\lambda; \hat{P}_{\text{test}})$.

The appropriate loss function involved in $R(\theta; P_{\text{model}})$ will be context-specific. Since we will be interested in binary classification, we will limit future discussion to the cross-entropy loss. How fairness should be defined is much more controversial. We will discuss various existing notions of fairness in the next section and advocate for a new causal measure of fairness that is especially adept at handling inherent biases in the dataset.

## 4 A NEW CAUSAL FAIRNESS MEASURE

Definitions of fairness can be largely divided into two camps. On one hand, we have non-causal fairness notions which typically operate by conditioning on the levels of the sensitive variable. For instance, enforcing equalised odds (Hardt et al., 2016) on a classifier $f_\theta$ amounts to enforcing two conditional distributions are the same, e.g. $p(f_\theta(\mathbf{x}) \mid \mathbf{a} = a, \mathbf{x} = x)$.

In the non-causal category, two fundamental definitions of fairness are demographic parity and con-ditional parity. The classifier $f_\theta(\mathbf{x})$ is said to exhibit **demographic parity** with the sensitive attribute $\mathbf{a}$ if $f_\theta(\mathbf{x}) \perp\!\!\!\perp \mathbf{a}$, where the shorthand $\perp\!\!\!\perp$ means independence. Intuitively, demographic parity as-sesses if the predicted score does not depend on the sensitive variable. For example, a classifier

predicting if a convicted criminal will re-offend exhibits demographic parity with respect to race if the distribution of $f_\theta(\mathbf{x})$ is the same irrespective of race. The drawbacks to demographic parity are well-documented (Hardt et al., 2016; Kleinberg, 2018). Essentially, when the base rates differ across values of the sensitive attribute, satisfying demographic parity can come at the cost of discrimination.

A more flexible framework of fairness is given by conditional parity, a term coined in Ritov et al. (2017). Let $\mathbf{u}$ be a random vector. The prediction score $f_\theta(\mathbf{x})$ is said to exhibit **conditional parity** with sensitive attribute $\mathbf{a}$ conditional on $\mathbf{u}$ if $f_\theta(\mathbf{x}) \perp\!\!\!\perp \mathbf{a} \mid \mathbf{u}$. Under the umbrella of conditional parity, Ritov et al. (2017) unified various fairness definitions. For instance, the notion of equalized odds, introduced in Hardt et al. (2016), is recovered by setting $\mathbf{u}$ to the true target class membership itself.

Intuitively, conditional parity asks for class predictions that are independent of the sensitive variable $\mathbf{a}$ conditioned on $\mathbf{u}$. For example, one could consider a classifier predicting if an applicant should be admitted to graduate school. Here, one may desire admission decisions to be independent of sex (demographic parity), or, for conditional parity, independent of sex *conditional* on a particular university department. That the notions of demographic and conditional parity can strongly differ and may lead seemingly paradoxical results was strikingly illustrated in Bickel et al. (1975) for graduate admissions at UC Berkeley.

### 4.1 Causal fairness in the overlap population

Taking the causal approach to defining fairness means replacing the question "Is the learning algorithm (conditionally) dependent on the sensitive attribute?" with the question "Does the sensitive attribute have a *causal effect* on the algorithm's predictions?" In an ideal world, we could intervene on the sensitive attribute by manipulating their values in an experiment and recording the outcomes. However, we usually only have access to observational data. Fortunately, causal inference tools can be used to glean causal effects from observational data. The tools are based on posing hypothetical questions about counterfactuals or potential outcomes: "What would have been the prediction outcome in a parallel universe where the only thing that changed about this subject was the value of the sensitive attribute?"

Our approach differs from previous works in causal fairness (Kusner et al., 2017; Kilbertus et al., 2017; Khademi et al., 2019) mainly in the causal estimand we use. Also, notably, we do not make use of structural equation models. A good review on causal approaches to algorithmic fairness can be found in Loftus et al. (2018).

Our new causal fairness notion is based on a special case of the weighted average treatment effect (WATE) (Hirano et al., 2003) whose development is motivated by the fact that in many situations due to selection bias, the study population may be different from the target population. To make valid causal statements, samples are weighed according to the covariate distributions of some target population.

We will use a special case of the WATE to measure and then penalise the causal link between the sensitive attribute and the intermediate representation in some layer of the neural network. Suppressing the dependence on the layer, let $\mathbf{h}$ denote values in the hidden layer. Adopting the potential outcome framework of Imbens & Rubin (2015) where we assume the Stable Unit Treatment Value Assumption, each intermediate representation $\mathbf{h}$ takes on one of two potential outcomes, $\mathbf{h}(0), \mathbf{h}(1)$ depending on whether $\mathbf{a} = 0$ or $\mathbf{a} = 1$. Note that $\mathbf{h} = \mathbf{a}\mathbf{h}(1) + (1 - \mathbf{a})\mathbf{h}(0)$, i.e. we can only ever observe one of the two potential outcomes.

Note that there are subtleties entailed by immutable variable such as race or gender. Kilbertus et al. (2017) argue for a workaround based on the idea of potential proxies. For instance the immutable characteristic of race has proxies such as name, visual features, languages spoken at home that can be conceivably manipulated.

Under *unconfoundedness*, i.e. $\mathbf{a}$ is independent of $\{\mathbf{h}(0), \mathbf{h}(1)\}$ conditional on $\mathbf{x}$, WATE is a class of causal estimands $\tau_g : \mathbb{R}^m \to \mathbb{R}^m$ parametrised by a function $g : \mathbb{R}^p \to \mathbb{R}$ given by

$$\tau_g(\mathbf{h}) = \frac{\mathbb{E}_{(\mathbf{x},\mathbf{a},\mathbf{y})\sim P_{\text{model}}}[g(\mathbf{x})(\mu_1(\mathbf{x}) - \mu_0(\mathbf{x}))]}{\mathbb{E}_{(\mathbf{x},\mathbf{a},\mathbf{y})\sim P_{\text{model}}}[g(\mathbf{x})]} \tag{4}$$

where $\mu_1(x) = \mathbb{E}(\mathbf{h}(1) \mid \mathbf{x} = x)$ and $\mu_0(x) = \mathbb{E}(\mathbf{h}(0) \mid \mathbf{x} = x)$. This form reveals WATE is indeed a measure of the causal effect in the target population specified by $g(x)$. Note when $g(x) = 1$ for all values of $x$, WATE reduces to the standard conditional average treatment effect (CATE), $\tau_{CATE} = \mathbb{E}_{(\mathbf{x},\mathbf{a},\mathbf{y}) \sim P_{\mathrm{model}}}(\mathbf{h}(1) - \mathbf{h}(0) \mid \mathbf{x} = x)$.

Henceforth, we focus our discussion on the case when $g(x) = e(x)(1 - e(x))$ where $e(x) = P(\mathbf{a} = 1 \mid \mathbf{x} = x)$ is also known as the propensity score. The propensity score is typically understood to be the probability of *treatment* given the covariate $\mathbf{x}$. (Recall in our case, the sensitive variable $\mathbf{a}$ plays the role of treatment.) The WATE corresponding to $g(x) = e(x)(1 - e(x))$ was called the *average treatment effect for the overlap population* (ATO) in Li et al. (2018a). The term overlap refers to the fact that the ATO articulates the causal effect among the **overlap population** which consists of subjects whose covariates could appear with substantial probability in either value of the sensitive attribute.

Let $\hat{e}(x)$ be the estimated propensity score function. In our experiments, we used a neural network to estimate the propensity score function. We actually also calibrated the predicted probabilties using the temperature scaling procedure of Guo et al. (2017).This neural net for learning the propensity score is trained once and for all on the training set.

A consistent estimator of the ATO (Li et al., 2018b) based on data $\{(\mathbf{x}_i, \mathbf{a}_i, \mathbf{h}_i)\}_{i=1}^n$ is

$$\hat{\tau}_{ATO}(\mathbf{h}) = \frac{\sum_{i=1}^n \mathbf{a}_i \mathbf{h}_i w_i}{\sum_{i=1}^n \mathbf{a}_i w_i} - \frac{\sum_{i=1}^n (1 - \mathbf{a}_i)\mathbf{h}_i w_i}{\sum_{i=1}^n (1 - \mathbf{a}_i)w_i} \tag{5}$$

where $w_i$ are the so-called overlap weights (Li et al., 2018a) given by

$$w_i = \begin{cases} 1 - \hat{e}(\mathbf{x}_i) & \text{if } \mathbf{a}_i = 1 \\ \hat{e}(\mathbf{x}_i) & \text{if } \mathbf{a}_i = 0. \end{cases}$$

Overlap weights derive their name from an emphasis on subjects with the most overlap in observed covariates $\mathbf{x}$ across the treatments (in our case the treatment is the sensitive attribute). Besides the nice interpretation of the ATO, there is also an important practical reason to adopt it as our causal estimand of choice. The overlap weights smoothly down-weigh subjects in the tails of the propensity score distribution, thereby mitigating the common problem of extreme propensity scores.

For precise conditions under which the estimator $\hat{\tau}_{ATO}$ is consistent, we refer the reader the set of regularity assumptions (called Assumption 1 to 5) in Hirano et al. (2003). These conditions are regulated to the distribution of $\mathbf{x}$ and distribution of $\mathbf{h}(0)$ and $\mathbf{h}(1)$. There is also a condition on the smoothness of the propensity score $e(x)$.

## 5 METHODOLOGY

In this section, we present a methodology for approximating the fairness-accuracy Pareto front of a fully-connected feedforward neural net classifier for the cross-entropy loss and the ATO fairness measure. The available data include a single binary sensitive variable $\mathbf{a}$, input variables $\mathbf{x} \in \mathbb{R}^p$, and binary response $\mathbf{y}$ indicating class membership. The input $\mathbf{x}$ is further standardised to mean 0 and variance 1. All discrete variables are dummy encoded.

Let $w^{(l)} \in \mathbb{R}^{m_l \times m_{l-1}}$ and $b^{(l)} \in \mathbb{R}^{m_l}$, $l = 1, \ldots, L$ be the parameters in the $l$-th layer of a fully-connected feedforward neural network with $L$ layers. Let $h^{(l)} : \mathbb{R}^{m_{l-1}} \to \mathbb{R}^{m_l}$ be the affine transformation

$$h^{(l)} = w^{(l)} v^{(l-1)} + b^{(l)}, l = 1, \ldots, L$$

where $v^{(0)} = id$ is the identity function and $m_0 = p$. The activation function $\sigma^{(l)} : \mathbb{R}^{m_l} \to \mathbb{R}^{m_l}$ is applied to obtain

$$v^{(l)} = \sigma^{(l)} \circ h^{(l)}, l = 1, \ldots, L.$$

The activation function in the final layer, $\sigma^{(L)}$, is restricted to the sigmoid function since we wish the classifier to output scores between 0 and 1. We use the ReLU activation function in all other layers for our experiments. Let $\mathbf{h}_i^{(l)}$ be shorthand for the application of the function $h^{(l)}$ to input feature $\mathbf{x}_i$, i.e. $\mathbf{h}_i^{(l)} = h^{(l)}(\mathbf{x}_i)$. Collect all parameters $w^{(l)}$ and $b^{(l)}$ for $l = 1, \ldots, L$ into the parameter vector $\theta$. The neural network is simply the function $f_\theta : \mathbb{R}^p \to [0, 1]$ given by $f_\theta(x) = v^{(L)}(x)$.

We will employ the binary cross-entropy loss $\mathcal{L} : [0,1] \times \{0,1\} \to \mathbb{R}$ given by $\mathcal{L}(\hat{y}, y) = y \log \hat{y} + (1-y) \log(1-\hat{y})$. Our interest is to determine, for the network $f_\theta$, the fairness-accuracy Pareto front associated to the (unknown) vector objective function

$$\begin{pmatrix} R(\theta; P_{\text{model}}) \\ U(\theta; P_{\text{model}}) \end{pmatrix} = \begin{pmatrix} -\mathbb{E}_{(\mathbf{x}, \mathbf{a}, \mathbf{y}) \sim P_{\text{model}}}(\mathbf{y} \log f_\theta(\mathbf{x}) + (1-\mathbf{y}) \log(1-f_\theta(\mathbf{x}))) \\ |\tau_{ATO}(f_\theta(\mathbf{x}))| \end{pmatrix}. \quad (6)$$

The first component measures classification error while the second component determines unfairness with respect to the ATO measure. (We would like both components to have low values.) To estimate the Pareto front of equation 6, we will use the strategy laid out in Section 3. Namely, we estimate each component of equation 6, scalarise the vector objective function using the Chebyshev method, and finally optimise the scalarised objective.

Estimation of $R(\theta; P_{\text{model}})$ is straightforward; we simply use the plug-in estimator $R(\theta; \hat{P}_{\text{data}}) = -\frac{1}{n} \sum_{i=1}^{n} [\mathbf{y}_i \log f_\theta(\mathbf{x}_i) + (1-\mathbf{y}_i) \log(1-f_\theta(\mathbf{x}_i))]$. Now we turn our attention to estimating $\tau_{ATO}((f_\theta(\mathbf{x}))$, the average effect of the sensitive attribute on the prediction for the overlap population. To achieve a low value for $\tau_{ATO}((f_\theta(\mathbf{x}))$, we could directly constrain the network to learn final predictions with low ATO. However, it may be preferable to penalise the ATO in the hidden layers of the network. This way, downstream analyses that involve transfer learning are also safeguarded against bias. See the exposition in Madras et al. (2018) for further benefits of learning fair internal representations. To keep the notation simple, let's say we penalise the hidden units in the some layer $l$. We then calculate the ATO in that layer to obtain

$$U(\theta; \hat{P}_{\text{data}}) = \left| \hat{\tau}_{ATO}(\mathbf{h}^{(l)}) \right| = \left| \frac{\sum_{i=1}^{n} \mathbf{a}_i \mathbf{h}_i^{(l)}(1 - \hat{e}(\mathbf{x}_i))}{\sum_{i=1}^{n} \mathbf{a}_i(1 - \hat{e}(\mathbf{x}_i))} - \frac{\sum_{i=1}^{n}(1 - \mathbf{a}_i)\mathbf{h}_i^{(l)}\hat{e}(\mathbf{x}_i)}{\sum_{i=1}^{n}(1 - \mathbf{a}_i)\hat{e}(\mathbf{x}_i)} \right|$$

We can generalise $U$ to penalise all layers in the neural network by summing over $l$. We will report on this experiment in Section 6.

Ideally, we would finely sample $\lambda$ in $[0,1]$ to find the set $\{\hat{\theta}_n^\lambda : \lambda \in [0,1]\}$ where $\hat{\theta}_n^\lambda$ is given by equation 3. But realistically, it might only be possible to perform the optimisation in equation 3 over a coarse grid of $\lambda$'s. It turns out that evenly distributed $\lambda$'s in the interval $[0,1]$ can often produce solutions that form clumps on the Pareto front, i.e. evenly distributed $\lambda$'s in $[0,1]$ do *not* produce evenly distributed points in the multi-objective space. Future work might seek to adaptively select the $\lambda$'s by implementing methods such as the Normal-Boundary-Interactive (Das & Dennis, 2000).

We also found it necessary to ensure the two terms in equation 6 are comparable in scale, so we standardised each term as follows. First, we ran the optimisation for $\lambda = 0$ and recorded the minimum $R_{min}$ and maximum $R_{max}$ of $R(\theta; \hat{P}_{\text{data}})$. Similarly we then ran the optimisation for $\lambda = 1$ to obtain $U_{min}$ and $U_{max}$. Then we standardised by $(R - R_{min})/(R_{max} - R_{min})$ for the expected loss and analogously for the unfairness measure.

Suppose we have available to us a testing set $\{(\mathbf{x}_i^*, \mathbf{a}_i^*, \mathbf{y}_i^*)\}_{i=1}^{m}$. We note that for evaluation, we assess $\begin{bmatrix} R(\hat{\theta}_n^\lambda, \hat{P}_{\text{test}}) \\ U(\hat{\theta}_n^\lambda, \hat{P}_{\text{test}}) \end{bmatrix}$ on the final prediction $f_{\hat{\theta}_n^\lambda}(\mathbf{x}_i^*)$ where

$$U(\hat{\theta}_n^\lambda, \hat{P}_{\text{test}}) = \left| \frac{\sum_{i=1}^{m} \mathbf{a}_i^* f_{\hat{\theta}_n^\lambda}(\mathbf{x}_i^*)(1 - \hat{e}(\mathbf{x}_i^*))}{\sum_{i=1}^{m} \mathbf{a}_i^*(1 - \hat{e}(\mathbf{x}_i^*))} - \frac{\sum_{i=1}^{m}(1 - \mathbf{a}_i^*)f_{\hat{\theta}_n^\lambda}(\mathbf{x}_i^*)\hat{e}(\mathbf{x}_i^*)}{\sum_{i=1}^{n}(1 - \mathbf{a}_i^*)\hat{e}(\mathbf{x}_i^*)} \right|.$$

## 6 EXPERIMENTS

In this section, we will apply the proposed methodology to two benchmarking datasets in the algorithmic fairness literature – the UCI adult income dataset and the ProPublica recidivism dataset. The two datasets are briefly summarised in Table 1 in Appendix A. Missing values were pre-processed according to the accompanying code. Our goals are as follows. In the UCI dataset, we wish to predict whether an individual has income above $50,000$ USD while remaining fair with respect to *gender*. Separately, we wish to perform the same prediction task while remaining fair with respect to *race*. In the recidivism dataset, we wish to predict whether an individual will recommit a crime in two years while remaining fair with respect to *race*.

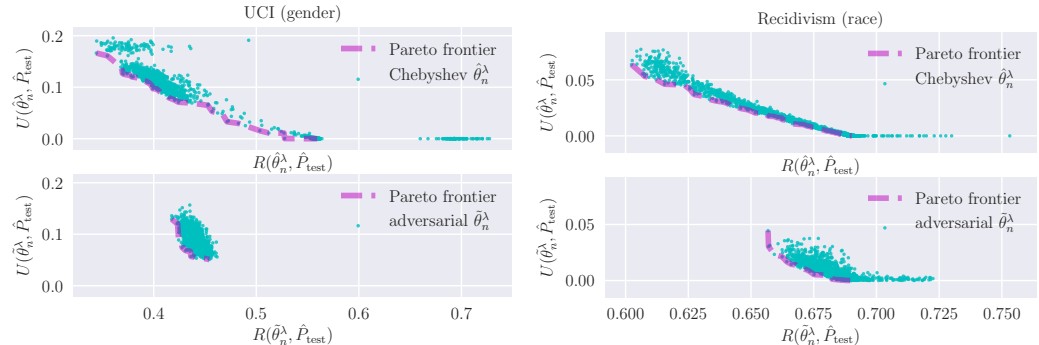

Figure 1: Each column of plots corresponds to a dataset and a sensitive attribute of interest. In all panels, we repeatedly split the data into training and testing sets, creating in total 100 sets of each. Then in each of the 100 training sets, for a collection of 15 $\lambda$'s in the interval $[0, 1]$, we find $\hat{\theta}_n^\lambda$ according to the Chebyshev scalarisation scheme (left panel) and $\tilde{\theta}_n^\lambda$ according to the adversarial approach (right panel). Thus there are a total of 1500 learned $\theta$'s in each plot and the magenta boundary is the Pareto frontier culled from these 1500 candidates. We can see the estimated Pareto front by the proposed methodology spans the fairness-accuracy space more than the adversarial approach.

We will achieve these goals by estimating the fairness-accuracy Pareto front of a binary classifier given by a fully-connected feedforward neural network. Note that we are conducting the analysis for the UCI dataset separately for race and gender. Future work should address fairness with respect to multiple sensitive attributes simultaneously; this would require an extension of the ATO to multiple "treatments" which was suggested as feasible future work in Li et al. (2018a).

Each dataset is split into a training set and a held-out test set, with the split reported in Table 1. First, the propensity scores are estimated using a neural net. Details of the propensity score network are given in Appendix A. For the neural net architecture defining $f_\theta$, the number of fully-connected layers and number of hidden nodes in each layer (held constant over the layers) were tuned for each dataset with the goal of not incurring over-fitting in the held-out test set. Each fully-connected layer is interspersed with a dropout layer with dropout probability $0.2$. The resulting architecture is reported in Table 2. The ReLU activation function is used in all intermediate layers while the sigmoid function is used in the output layer.

To learn the network, we use the ADAM optimisation algorithm (Kingma & Ba, 2014). The initial learning is fixed at $0.001$. We reduce the learning rate when the training loss has stopped decreasing by using the `ReduceLROnPlateau` scheduler in PyTorch, setting the factor and patience variables to $0.9$ and $10$, respectively. All training took place over 500 epochs. Mini-batch size was chosen to be around $5\%$ of the training set size; 150 and 1000 minibatch sizes were used in the recidivism and UCI datasets, respectively.

For a given dataset and a sensitive attribute of interest, we repeatedly split the data into training and testing sets, creating in total 100 sets of each. Then, in each of the 100 training sets, we find $\hat{\theta}_n^\lambda$, according to equation 3, for a collection of 15 $\lambda$'s in the interval $[0, 1]$. The quality of the approximation is assessed using the corresponding test set. Thus we produce a total of 1500 learned network parameters and each one can be plotted in the fairness-accuracy space. Each column of Figure 1 shows 1500 $\hat{\theta}_n^\lambda$'s as well as the Pareto front culled from these 1500 Pareto candidates. The culling simply checks which of the 1500 points are non-dominated points. As we sweep from the top-left corner to the bottom-right corner, we move from neural networks exhibiting high-accuracy-low-fairness to those exhibiting low-accuracy-high-fairness.

Next, we investigate the visual effect of dialling $\lambda$ between 0 and 1. In Figure 2, we display the distribution of the classifier's prediction in the recidivism dataset broken down by class label and sensitive attribute. Each panel of Figure 2 is a different $\lambda$ value. In addition to reporting the ATO measure of fairness, we also indicate other non-causal fairness metrics including Equalised Odds, Equal Opportunity, and Demographic Parity. Similar visualisation for the UCI (gender) and UCI (race) dataset can be found in Figures 3 and 4 in the Appendix, respectively.

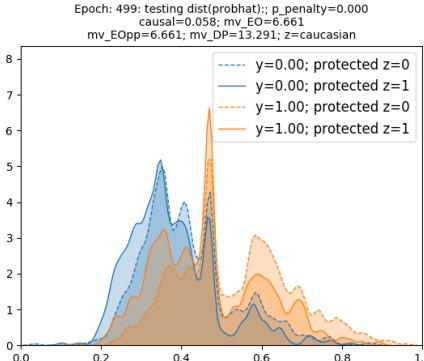 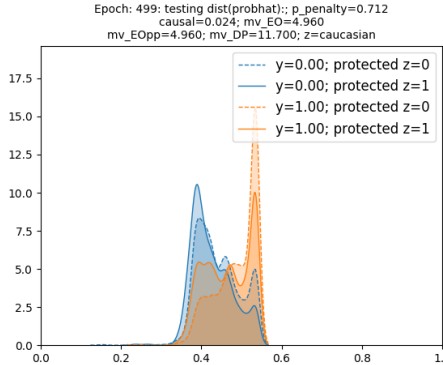

Figure 2: Distributions of the predicted probabilities in the test set of the recidivism dataset are plotted for $\lambda = 0$ (left) and $\lambda = 0.712$ (right). The distributions are broken down by different values of the true target label $\mathbf{y}$ and value of the sensitive attribute $\mathbf{z}$. Besides the causal fairness notion we introduced in this work, we also indicate equalised odds (mv_EO), equality of opportunity (mv_EOpp) and demographic parity (mv_DP).

We also repeated this experiment by modifying the fairness measure to penalise intermediate representations in *all* layers, i.e. set $U(\theta; \hat{P}_{\text{data}}) = \sum_{i=1}^{L} \left| \hat{\tau}_{ATO}(\mathbf{h}^{(l)}) \right|$, for which the results are reported in Figure 6 in Appendix A. It seems that penalising all layers makes training the neural network more difficult as we encounter many iterations where we end up in a local optimum. For simplicity, we had used the same network architecture as in the case of penalising one internal layer which may have prompted the performance issue.

**Comparison to alternatives**   We did not find other works in the algorithmic fairness literature that address the specific task of finding the fairness-accuracy Pareto front of a neural network. Given this, we instead looked for methods where there was some type of tuning parameter that controls the trade-off between fairness and accuracy. By dialling this tuning parameter, one could hope to sweep out a set of classifiers that live in different parts of the fairness-accuracy landscape.

Given the diversity of fairness methods, due in part to the fairness definition used, we decided to implement the adversarial training technique proposed in Louppe et al. (2017) which is not based on a specific fairness criterion. The idea is intuitive; the classifier and adversarial are engaged in a zero-sum game. The classifier network, call its parameters $\theta_{clf}$, attempts to make the best binary classification. The adversary, on the other hand, attempts to make the best prediction of the binary sensitive attribute based on the classifier's prediction. Let $\theta_{adv}$ denote the parameters of the adversarial network. The overall objective is $\tilde{\theta}_n^\lambda = \arg\min_{\theta_{clf}} \left[ Loss_\mathbf{y}(\theta_{clf}) - \lambda Loss_\mathbf{a}(\theta_{clf}, \theta_{adv}) \right]$. where the first loss measures the prediction of $\mathbf{y}$ and the second loss measures the prediction of the sensitive attribute $\mathbf{a}$. Both losses were chosen to be the binary cross-entropy loss.

Our implementation of Louppe et al. (2017) is based on GoDataDriven's post on fairness in machine learning with adversarial networks. Following their choice of epochs, we alternate the following steps over 200 epochs: (1) train the adversarial network for a *single epoch*, holding the classifier network fixed and (2) train the classifier network on a *single sampled mini batch*, holding the adversarial network fixed. Details on the adversarial network architecture are provided in Appendix A. For the classifier network, we employed the same network as above in our proposed methodology and kept all training choices, such as the optimisation algorithm and mini-batch size, the same. The classifier was pretrained for 2 epochs.

We modified GoDataDriven's code so that we can sweep $\tilde{\theta}_n^\lambda$ over the parameter $\lambda$. The result of the adversarial approach is shown in the right column of Figure 1. Once again, using the same 100 training and testing pairs as above, we find solutions to $\tilde{\theta}_n^\lambda$ for a set of 15 $\lambda$'s in $[0, 1]$, for a total of 1500 Pareto candidates. We can immediately see that compared to the proposed methodology, the adversarial is less capable of finding a Pareto front that spans the fairness-accuracy space. Indeed the set of dominated (non-dominated) points in the adversarial approach is larger (smaller) relative to the proposed approach, see Table 3.

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

## A APPENDIX

In this section, we provide additional details on the experiments conducted in Section 6. First, in Table 1, we summarise the datasets on which all experiments were conducted. The feedforward architecture used in the proposed methodology is given in Table 2 for each of the three data settings.

Next, we describe the neural net we employed to estimate the propensity scores $P(\mathbf{a} = 1 \mid X)$. We actually used the same neural net for all three data settings in Table 1. Each of the 3 fully-connected layer, with 32 hidden units each, is interspersed with a dropout layer with dropout probability 0.2. The ReLU activation function is used in all intermediate layers while the sigmoid function is used in the output layer. The cross-entropy loss is used between the estimated scores and the true labels dictated by $\mathbf{a}$. To learn the network, we use the ADAM optimisation algorithm (Kingma & Ba, 2014). The initial learning is fixed throughout at 0.001. Training took place over 100 epochs. Mini-batch size was chosen to be around $5\%$ of the training set size; 150 and 1000 minibatch sizes were used in the recidivism and UCI datasets, respectively. After the propensity neural network is trained, we apply the calibration technique proposed in Guo et al. (2017) to calibrate the probability predictions. We used their GitHub code with no modification.

Figure 5 shows the additional output from the experiment that produced Figure 1 on the UCI dataset with race as the sensitive attribute. We also repeated the experiment in Section 6 by penalising the ATO in all layers. The results are shown in Figure 6. The adversarial approach we compared the proposed methodology against used a network with 4 hidden layers with 32 hidden units in each. ReLU activations were used throughout except in the final layer where the sigmoid function is used. The adversarial network was pretrained for 5 epochs. Optimisation used ADAM and minibatch sizes described in Table 1.

Table 1: Dataset descriptions

| | | dataset features | | | | |
|---|---|---|---|---|---|---|
| Dataset | $dim(\mathbf{x})$ | binary outcome $y$ | sensitive $\mathbf{a}$ | training size | testing size | minibatch size |
| Recidivism | 12 | Reoffend in 2 years? | binary race | 3086 | 3086 | 150 |
| UCI | 93 | Income above 50K? | binary race | 15470 | 15470 | 1000 |
| UCI | 93 | Income above 50K? | binary gender | 15470 | 15470 | 1000 |

Table 2: Network architecture

| | neural network features | |
| Dataset | layers $L$ | hidden nodes |
|---|---|---|
| Recidivism | 4 | 4 |
| UCI | 32 | 10 |
| UCI | 32 | 10 |

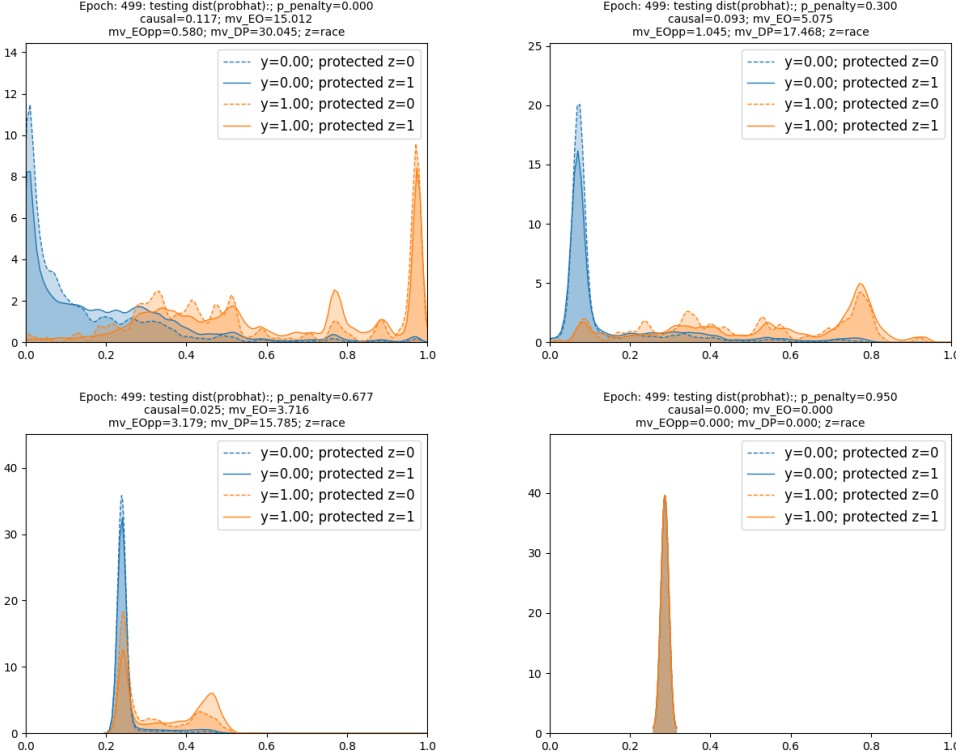

Figure 3: Prediction probabilities in the test set of UCI (race) for varying values of $\lambda$, indicated by p_penalty in the heading of each plot.

Table 3: A comparison between the proposed methodology the adversarial technique for finding the Pareto front in each of the three data settings of Figure 1. Reported are the number of *non-dominated* points. Higher is better.

| | UCI (gender) | UCI (race) | Recidivism |
|---|---|---|---|
| Proposed | 44 | 33 | 89 |
| Adversarial | 13 | 9 | 27 |

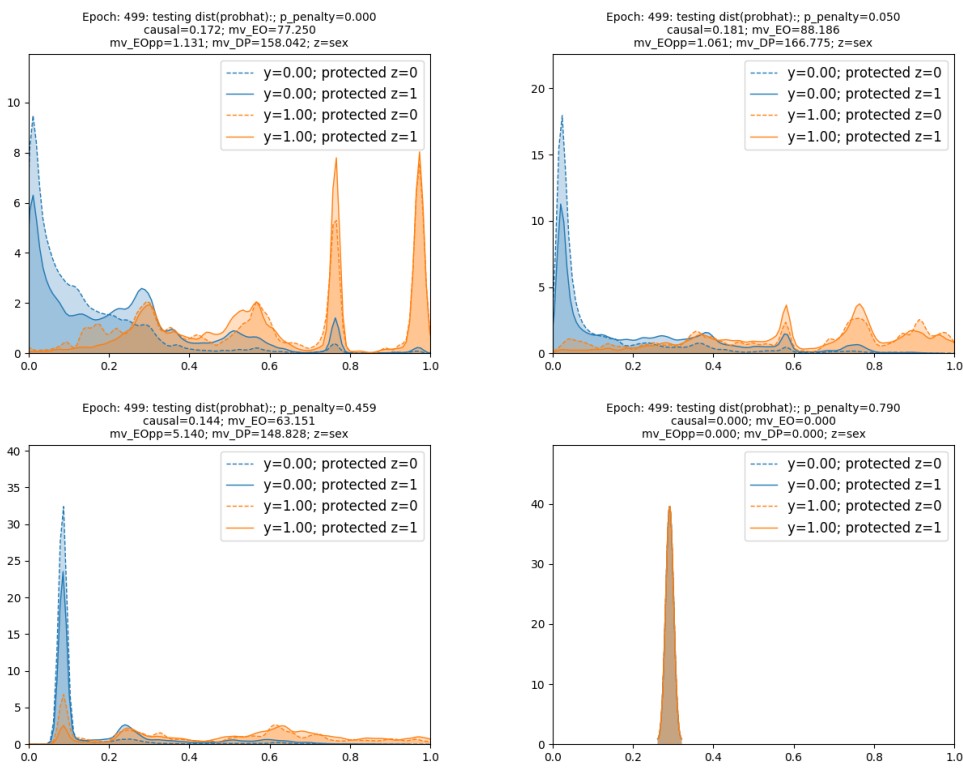

Figure 4: Prediction probabilities in the test set of UCI (gender) for varying values of $\lambda$, indicated by p_penalty in the heading of each plot.

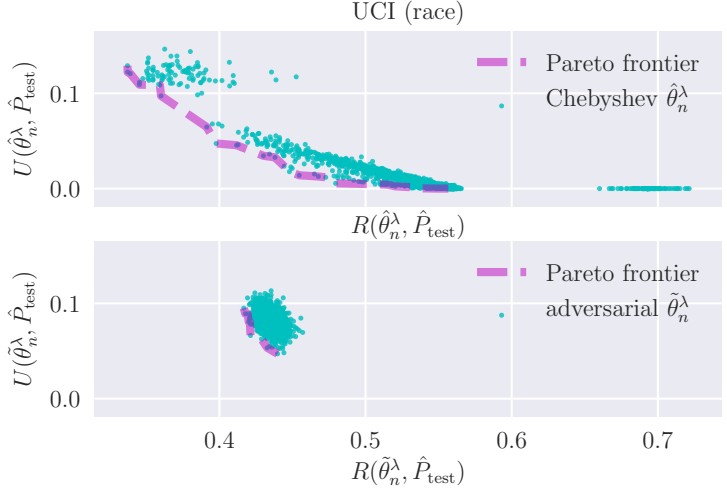

Figure 5: This reports the result for UCI (race) in the same experiment that produced Figure 1.

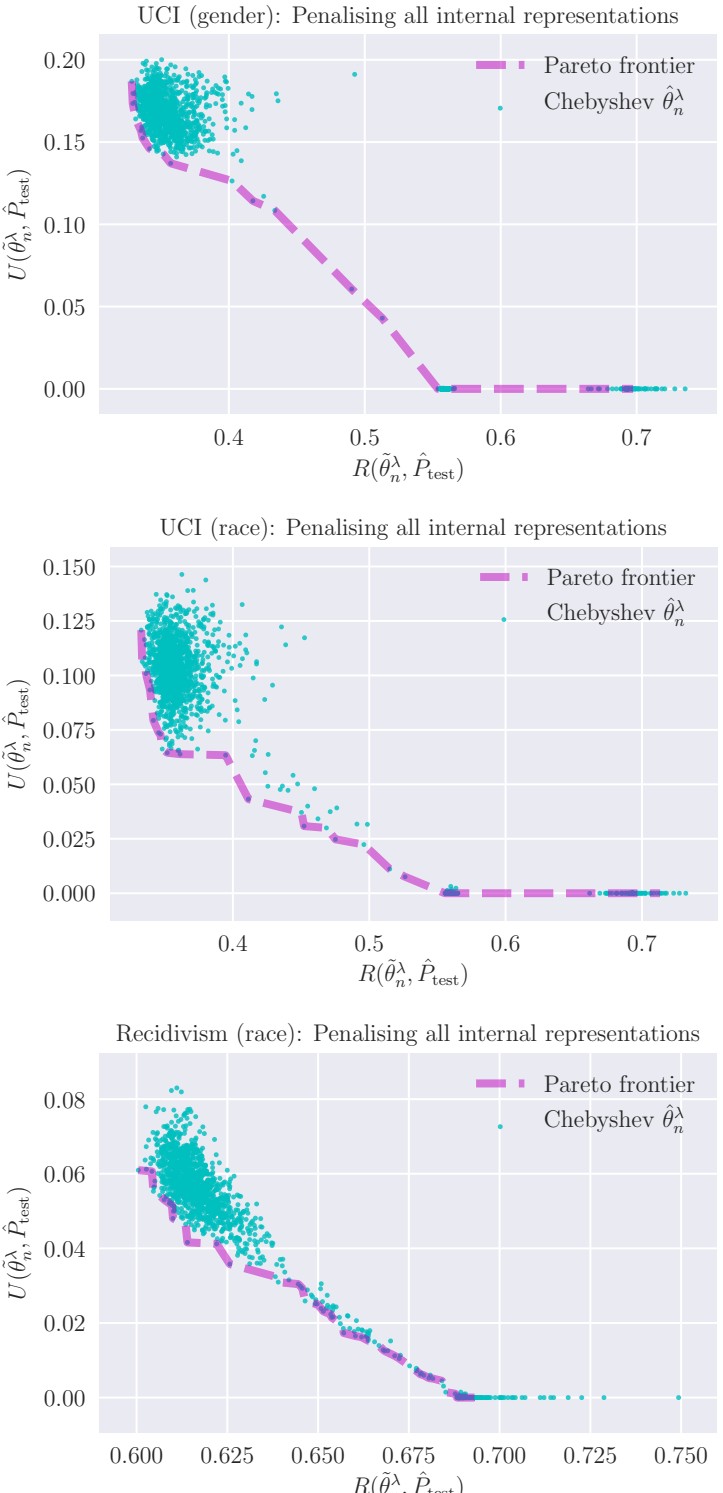

Figure 6: We repeated the experiment in Figure 1 changing only the fairness penalty to penalise the ATO in *all* layers. There is a drop in the quality of the Pareto front estimation compared to penalising just one internal layer. Namely, more of the candidate points are dominated points in this modified experiment where we've penalised ATO in all layers. It seems that we should have perhaps also tuned for a brand new architecture given this new penalty.

