# OpenReview forum: "The fairness-accuracy landscape of neural classifiers"
_ICLR.cc/2020/Conference — Reject_

### Official Review · AnonReviewer3 · 2019-10-22
**Official Blind Review #3**

**Rating:** 1

**Review:**

This paper proposes a method to approximate the "fairness-accuracy landscape" of classifiers based on neural networks.
The key idea is to set up a multi-dimensional objective, where one dimension is about prediction accuracy and another
about fairness. The fairness component relies on a  definition of fairness based on causal inference, relying on the
idea that a sensitive attribute should not causally affect model predictions.

I found the causal idea intriguing, since it makes sense that we don't want a sensitive attribute to have a causal effect.
However, there may be several problems with this approach:

1) For a causal estimate to be valid we need several assumptions. For example, we need A (the sensitive attribute)
to be independent of potential outcomes conditional on X --- the so-called "unconfoundedness assumption" in causal inference.
We also need "positivity", i.e., that 0< P(A=1|X) <1.
These assumptions are not discussed in the paper. Furthermore, the particular context of the paper, where the treatment is actually an immutable characteristic, makes such discussion much more subtle.
What will we do, for instance, if there are no A=1 in the sample when X = ...?


2) The authors seem to assume that the propensity score model is well specified. This can be tested, e.g., using [1].
What do we do when this fails?


3) Why do we want U to be small, i.e., why do we want the causal effect of A to be small, is never justified.
In particular, its relation to "fairness" is never fleshed out, but just assumed to be so.
This can be problematic when, say, we are missing certain important X that are important for A.
Then, there will be a measurable causal effect of A on h().

Some other problems:
- What is the reason for focusing on 'neural classifiers'? There is nothing specific in the method or analysis
that relates to neural networks, except for the use of the causal estimand in a 'hidden layer'.

- In the Introduction, the authors could cite the works of Amartya Sea, etc., on fairness.
Certainly the study of fairness problems did not start in 2016.

- What exactly is a "sensitive attribute"? If we don't want to bias our predictions, then why include it in the analysis?

- It is unclear what is new and what is related work in page 3.

- Sec. 4: The claim that "causal inference is about situations where manipulation is impossible" discards
voluminous work in causal inference through randomized experiments. In fact, many scientists would
agree that causal inference is impossible without manipulation.

- As mentioned above, why this particular estimand leads to more "fairness" is never explained.

- Do we need a square or abs value in Eq (5)?

- The experimental section is weak and does not illustrate the claims well.
   It would be important to explain the choice of the particular causal estimand, the choice of the hidden layer to put the estimand in, to explore the choice of the objective, and so on. Currently, none of these choices/design aspects are being investigated.



[1] "A specification test for the propensity score using its distribution conditional
on participation" (Shaikh et al, 2009)

**Experience Assessment:**

I have published one or two papers in this area.

**Review Assessment: Checking Correctness Of Derivations And Theory:**

I carefully checked the derivations and theory.

**Review Assessment: Checking Correctness Of Experiments:**

I carefully checked the experiments.

**Review Assessment: Thoroughness In Paper Reading:**

I read the paper thoroughly.

---

> ### Author Response · Authors · 2019-11-15
> **Response to Review #3 (Part 1 of 3)**
>
> The reviewer is correct that we failed to make the assumptions regarding the causal estimand explicit. These necessary assumptions are now clearly stated in the revision:
> - In adopting the potential outcome framework of Imbens and Rubin 2015, we assume the Stable Unit Treatment Value Assumption
> - Under unconfoundedness, i.e.\ $A$ is independent of $\{h(0),h(1)\}$ conditional on $X$, WATE is a class of causal estimands that includes the ATO as a special case
> - In order for the ATO estimate to be consistent, we refer the reader to the set of regularity assumptions (called Assumption 1 to 5) in Hirano et. al 2003. A few of these conditions are regulated to the distribution of $X$ and distribution of $h(0)$ and $h(1)$. There is also a condition on the smoothness of the propensity score $e(x)$ which is even stricter than positivity.
>
> The reviewer is correct that we should’ve done a better job discussing the subtlety involved when the treatment is actually an immutable characteristic. We have added a brief discussion in the revision that echos similar concerns raised in Kilbertus et. al 2017. Namely, an explicit distinction should be drawn between the sensitive attribute (for which interventions are often impossible in practice) and its proxies. For instance the immutable characteristic of race has proxies such as name, visual features, languages spoken at home that can be conceivably manipulated.
>
> Next, the positivity assumption can be checked in the sample, i.e. by checking whether there are observational units that are “treated” ($A=1$) and “untreated” ($A=0$) in each stratum of $X$. If we observe a stratum of $X$ in which there are only treated or only untreated, we need to ask ourselves if this is happening by pure chance due to sampling variability or this is happening because of some structural reason (units with covariates in this stratum are deterministically always “treated” or always “untreated”). The latter is very hard to deal with whereas the former is not, strictly speaking, a violation of the positivity assumption. But nonetheless scarcity of data in certain strata of $X$ does pose a practical issue in identifying the causal effect. This has been studied in the causal inference literature and we may implement some of the suggestions in [2] such as restriction of the data to those observational units who do not violate the positivity assumption or excluding certain covariates responsible for positivity violations.
>
> Regarding the specification of the propensity score model, we echo the viewpoint in Li. et al 2018 that for the purposes of estimating the ATO, a good propensity score model is one that leads to covariate balance in the sample, not one that allows us to make inferences about treatment assignment probabilities in the population. Thus it would seem that we can perhaps get away with a misspecified propensity score model as long as it achieves covariate balance in our sample.
>
> To answer the reviewer's question about $U$, we want $U$, the causal effect of the sensitive attribute $A$, to be small because we don't want a sensitive attribute to have a causal effect on the outcome. The possibility of confounding by unobserved variables is of course a real concern; it is part of what makes causal inference such a challenging task. To really deal with this type of problem, domain experts and stakeholders have to be involved to think about how the data was gathered.
>
> [2] “Diagnosing and responding to violations in the positivity assumption” (Petersen et. al 2012)

---

> > ### Author Response · Authors · 2019-11-15
> > **Response to Review #3 (Part 2 of 3)**
> >
> > Next, we respond to the rest of the comments in Review #3, point-by-point.
> >
> > - What is the reason for focusing on 'neural classifiers'?...
> >
> > Indeed, the fairness-accuracy Pareto front can also be estimated for other classifiers. We chose to focus on neural networks because they represent the state-of-the-art in classification approaches these days. Also, at the outset, it wasn’t immediately obvious that we could use multiobjective optimisation techniques to efficiently find non-dominanted points of a neural network. While our approach can also be useful for non-neural network classifiers we show here that the proposed approach easily integrates into a neural network setup and in particular allows removing the influence of sensitive attributes on all layers of a neural network.
> >
> > - In the Introduction, the authors could cite the works of Amartya Sea, etc., on fairness...
> >
> > We regret not being more thorough in our references. We now cite the work of Sea on fairness in our revision.
> >
> > - What exactly is a "sensitive attribute"? If we don't want to bias our predictions, then why include it in the analysis?
> >
> > The sensitive attribute is the attribute we want the algorithm to be unbiased with regards to, as much as possible. We need access to it during training so that we can achieve debiasing. However, importantly, at deployment time, we do not need access to the sensitive attribute to make a fair classification decision. Note that it is well understood that simply removing the sensitive attribute from the entire training process does not promote a fair classifier because there may be other variables highly correlated with the sensitive attribute that the algorithm can still leverage.
> >
> > - It is unclear what is new and what is related work in page 3.
> >
> > The top of page 3 describes the Pareto front and scalarisation schemes for estimating it which is based on well established concepts in multiobjective optimisation. The bottom of page 3 describes the estimation of the Pareto front specific to our supervised learning setup which is new.
> >
> > - Sec. 4: The claim that "causal inference is about situations where manipulation is impossible"...
> >
> > We apologise for the confusing way in which we stated this. We have removed the sentence. We were trying to say that in an ideal world, we could intervene on the sensitive attribute by manipulating their values in an experiment and recording the outcomes. However, we usually only have access to observational data. Fortunately, causal inference tools can be used to glean causal effects from observational data.
> >
> > - As mentioned above, why this particular estimand leads to more "fairness" is never explained.
> >
> > Because we have defined fairness to mean the sensitive attribute has no causal effect on the classification, this means we want the ATO causal estimand to be low.
> >
> > - Do we need a square or abs value in Eq (5)?
> >
> > Yes, indeed. Thank you for pointing out this typo which we’ve fixed in the revision.
> >
> > - The experimental section is weak and does not illustrate the claims well.
> >
> > We acknowledge the limitations of our current experimental section. For the final version of the paper, we will apply our proposed methodology on five other benchmarking datasets provided in the AI Fairness 360 toolkit.
> >
> > In the meantime, we added some additional visualisation in the experimental section which shows the visual effect of dialling $\lambda$ between 0 and 1. Namely, for several values of the penalty parameter $\lambda$, we plot the distribution of the final prediction broken down by true class membership $Y$ and sensitive attribute $A$. In addition to reporting the ATO measure of fairness, we also indicate other non-causal fairness metrics including Equalised Odds, Equal Opportunity, and Demographic Parity.

---

> > > ### Author Response · Authors · 2019-11-15
> > > **Response to Review #3 (Part 3 of 3)**
> > >
> > > In response to "It would be important to explain the choice of the particular causal estimand, the choice of the hidden layer to put the estimand in, to explore the choice of the objective, and so on. Currently, none of these choices/design aspects are being investigated."
> > >
> > > In the revision we have carefully discussed each of these issues. Regarding the choice of the particular causal estimand, the ATO has the nice interpretation of focusing on subjects with the most overlap in observed covariates. There is also an important practical reason to adopt it as our causal estimand of choice. The overlap weights smoothly down-weigh subjects in the tails of the propensity score distribution, thereby mitigating the common problem of extreme propensity scores.
> > >
> > > As for the choice of the hidden layer to put the causal constraint in, we experimented with penalising just one of the internal layers versus penalising all internal layers. The experimental results for the latter are placed in the Appendix. We see that although penalising all layers has the benefit of allowing downstream transfer learning tasks to be fair, the training process encounters more convergence issues as can be seen from Figure 6 in the appendix. We are investigating an approach where we penalise layer by layer so that the training has a better chance of converging.
> > >
> > > Finally, regarding the choice of the objective, we suppose the reviewer means the choices of the vector objective function in Equation 1? In that case, we think it is important to look at the vector objective because both accuracy and fairness are desirable in the learning algorithm. Our particular choice of the expected cross-entropy loss for measuring fairness is common in classification settings. Our choice of using the ATO for fairness is again because we think a causal estimand could reveal insights that measures like conditional parity cannot. Furthermore, we choose ATO to be the causal estimand because it has a nice interpretation and does not suffer from extreme propensity scores.

---

### Official Review · AnonReviewer2 · 2019-10-25
**Official Blind Review #2**

**Rating:** 6

**Review:**

The authors propose a novel joint optimisation framework that attempts to optimally trade-off between accuracy and fairness objectives, since in its general formal counterfactual fairness is at odds with classical accuracy objective. To this end, the authors propose optimising a Pareto front of the fairness-accuracy trade-off space and show that their resulting method outperforms an adversarial approach in finding non-dominated pairs.

As main contributions, the paper provides:
* A Pareto objective formulation of the accuracy fairness trade-off
* A new causal fairness objective based on the existing Weighted Average Treatment Effect (WATE) and Average Treatment Effect for the Overlap Population (ATO)

Overall, I think the paper makes an interesting contribution to the field of fairness and that the resulting method seems quite attractive for a real-world practitioners. However, I found the writing / notation imprecise at times and the experimental section too small (lacking an extensive set of baselines, and only on two datasets). For these reasons, I give it a Weak Accept.

Some feedback on notation / writing:
* Typo on page 2, the loss L should be defined on X x Y and not Y x Y
* In page 5, h is being used without being introduced first
* the justification for using ATO in the internal layers of the network is a bit insufficient

In terms of suggestions, I think the experimental section needs to be extended and that the various modelling choices need to be explored and/or be further justified.

**Experience Assessment:**

I have read many papers in this area.

**Review Assessment: Checking Correctness Of Derivations And Theory:**

I assessed the sensibility of the derivations and theory.

**Review Assessment: Checking Correctness Of Experiments:**

I assessed the sensibility of the experiments.

**Review Assessment: Thoroughness In Paper Reading:**

I read the paper at least twice and used my best judgement in assessing the paper.

---

> ### Author Response · Authors · 2019-11-15
> **Response to Review #2**
>
> We thank the reviewer for their careful reading and feedback. We have combed through our original submission to fix imprecision in writing and notation, including the specific points raised above. (Actually, regarding the loss, this is not a typo but really what we mean…).
>
> We have also added better explanation of why we penalise the average treatment effect for the overlap population (ATO) in the internal layers. Basically, we believe that the best safeguard against unfairness in a neural net classifier is to constrain the network to learn fair intermediate representations. This is because internal representations of neural networks are commonly assumed to contain useful information and may be subsequently employed in transfer learning. Therefore it would be important to constrain internal layers of the neural network to be fair as well. Our experimental results include a setup where all intermediate layers are penalised and a setup where only the next-to-last layer is penalised. The former makes the training more difficult although the estimated Pareto front is still reasonable. We will investigate in future work how to train this setup in a better way. Nonetheless in both setups it is interesting that only constraining intermediate representations to be fair is sufficient to obtain fairness on the final prediction.
>
> We acknowledge the limitations of our current experimental section. We recently became aware of the AI Fairness 360 Tool, a Python package that includes a convenient interface to seven popular datasets in the fairness literature. In the original submission we analysed two of the datasets contained therein — the Adult Census Income and the ProPublica Recidivism dataset. Unfortunately there is not enough time during this discussion phase to run our proposed methodology on the other five datasets provided in AI Fairness 360, but we will do this for the final version of the paper.
>
> In the meantime, we were able to add some additional visualisation (Figures 2-4) in the experimental section which shows the visual effect of dialling $\lambda$ between 0 and 1. Namely, for several values of the penalty parameter $\lambda$, we plot the distribution of the final prediction broken down by true class membership $Y$ and sensitive attribute $A$. In addition to reporting the ATO measure of fairness, we also indicate other non-causal fairness metrics including Equalised Odds, Equal Opportunity, and Demographic Parity.

---

### Official Review · AnonReviewer1 · 2019-10-28
**Official Blind Review #1**

**Rating:** 3

**Review:**

General:
The paper proposed to use a causal fairness metric, then tries to identify the Pareto optimal front for the vectorized output, [accuracy, fairness]. While the proposed method makes sense, I am not sure what exactly their contribution is. It is kind of clear that Pareto optimal exists, and what they did is to run the experiments multiple times with multiple \lambda values for the Chebyshev method and plot the Pareto optimal front.

Pro:
Ran multiple experiments and drew the Pareto optimal front for the considered dataset.

Con & Question:
The so-called causal fairness metric does not seem to be any more fundametal than the other proposed metrics. It seems like they worked with another metric.
After defining the fairness metric, everything else seems straightforward. Just use test (validation) set to estimate the accuracy & fairness, then plot the results on the 2d plane.
Can we identify the Pareto optimal front without running all 1500 experiments? What happens when running a model takes long to train? Then, the proposed method cannot be practical.
Figure fonts are very small and hard to see.

**Experience Assessment:**

I have read many papers in this area.

**Review Assessment: Checking Correctness Of Derivations And Theory:**

I carefully checked the derivations and theory.

**Review Assessment: Checking Correctness Of Experiments:**

I assessed the sensibility of the experiments.

**Review Assessment: Thoroughness In Paper Reading:**

I read the paper thoroughly.

---

> ### Author Response · Authors · 2019-11-15
> **Response to Review #1**
>
> We thank the reviewer for the opportunity to clarify the paper’s contributions:
> - This work is among the first in algorithmic fairness to focus on the fairness-accuracy tradeoff curve. Formulating the trade-off curve as a Pareto front estimation problem, we demonstrate that it is indeed possible to find a significant set of non-dominated points for a neural network, which is not immediately obvious given how difficult it is to train even a scalar objective.
> - The generality of the proposed methodology framework allows end-users to supply their own fairness and accuracy measures.
> - This work also investigates a new causal measure for the purpose of assessing algorithmic fairness based on the average treatment effect for the overlap population (ATO) proposed in Li et. al (2018) which can achieve covariate balance and does not suffer from extreme propensity scores.
> - The proposed methodology can achieve fairness on the final prediction even though it only constrains intermediate representations of the neural network to be fair. This approach may have benefits for downstream transfer learning tasks.
>
> In the original submission, we had discussed, arguably, the two most fundamental fairness concepts in the existing literature — demographic parity and conditional parity. The latter envelops several existing fairness metrics, e.g. the concept of equalised odds introduced by Hardt et al. (2016) is an instance of conditional parity. Both concepts are based on the joint distribution of the classifier, the sensitive attribute $A$, the covariate $X$, and the outcome $Y$. This opens the door for using a wide variety of statistical tools to estimate these quantities. Unfortunately as documented by works such as Kilbertus et. al 2017, these approaches are purely observational in nature and cannot distinguish subtle scenarios in which the joint distributions are the same but there is clear unfairness.
>
> For these reasons, we believe causal notion of fairness might provide fresh insights. Our idea is that when the dataset is itself collected under unfair practices, we must correct for the covariate imbalance before assessing fairness. We chose to employ the ATO proposed in Li et. al 2018 because it avoids the instability of weights resulting from extreme propensity scores.
>
> Regarding the reviewer's concern about run time, indeed an attempt at identifying the Pareto front can certainly be made by running fewer experiments, but because convergence issues are commonly encountered during the training of a neural network, the quality of the estimated front will likely suffer.
>
> We understand the reviewer’s concern that the proposed method will be cumbersome to implement if a single iteration takes very long to train. Fortunately, there are more sophisticated methods for selecting the trade-off parameter ($\lambda$) in the multi-objective optimisation literature such as the Normal Boundary Interactive method in Das and Dennis 1997. We have indicated in the paper that we plan to explore such techniques in future work so that a Pareto front can be accurately identified in a more efficient manner.”
>
> Finally, regarding the figure font size, we have fixed this issue in the revision.

---

### Decision · Program_Chairs · 2019-12-19

**Decision:**

Reject

**Comment:**

This manuscript investigates and characterizes the tradeoff between fairness and accuracy in neural network models. The primary empirical contribution is to investigate this tradeoff for a variety of datasets.

The reviewers and AC agree that the problem studied is timely and interesting. However, this manuscript also received quite divergent reviews, resulting from differences in opinion about the novelty of the results. IN particular, it is not clear that the idea of a fairness/performance tradeoff is a new one. In reviews and discussion, the reviewers also noted issues with clarity of the presentation. In the opinion of the AC, the manuscript is not appropriate for publication in its current state.